# The Correlation between Subolesin-Reactive Epitopes and Vaccine Efficacy

**DOI:** 10.3390/vaccines10081327

**Published:** 2022-08-16

**Authors:** Marinela Contreras, Paul D. Kasaija, Fredrick Kabi, Swidiq Mugerwa, José De la Fuente

**Affiliations:** 1SaBio, Instituto de Investigación en Recursos Cinegéticos (IREC), Consejo Superior de Investigaciones Científicas (CSIC), Universidad de Castilla-La Mancha (UCLM)-Junta de Comunidades de Castilla-La Mancha (JCCM), Ronda de Toledo s/n, 13005 Ciudad Real, Spain; 2National Livestock Resources Research Institute (NaLIRRI/NARO), Kampala P.O. Box 5704, Uganda; 3Center for Veterinary Health Sciences, Department of Veterinary Pathobiology, Oklahoma State University, Stillwater, OK 74078, USA

**Keywords:** subolesin, Q38, epitope, vaccine, cattle, quantum vaccinomics, tick

## Abstract

Vaccination is an environmentally-friendly alternative for tick control. The tick antigen Subolesin (SUB) has shown protection in vaccines for the control of multiple tick species in cattle. Additionally, recent approaches in quantum vaccinomics have predicted SUB-protective epitopes and the peptide sequences involved in protein–protein interactions in this tick antigen. Therefore, the identification of B-cell–reactive epitopes by epitope mapping using a SUB peptide array could be essential as a novel strategy for vaccine development. Subolesin can be used as a model to evaluate the effectiveness of these approaches for the identification of protective epitopes related to vaccine protection and efficacy. In this study, the mapping of B-cell linear epitopes of SUB from three different tick species common in Uganda (*Rhipicephalus appendiculatus, R. decoloratus*, and *Amblyomma variegatum*) was conducted using serum samples from two cattle breeds immunized with SUB-based vaccines. The results showed that in cattle immunized with SUB from *R. appendiculatus* (SUBra) all the reactive peptides (Z-score > 2) recognized by IgG were also significant (Z-ratio > 1.96) when compared to the control group. Additionally, some of the reactive peptides recognized by IgG from the control group were also recognized in SUB cocktail–immunized groups. As a significant result, cattle groups that showed the highest vaccine efficacy were *Bos indicus* immunized with a SUB cocktail (92%), and crossbred cattle were immunized with SUBra (90%) against *R. appendiculatus* ticks; the IgG from these groups recognized overlapping epitopes from the peptide SPTGLSPGLSPVRDQPLFTFRQVGLICERMMKERESQIRDEYDHVLSAKLAEQYDTFVKFTYDQKRFEGATPSYLS (Z-ratio > 1.96), which partially corresponded to a Q38 peptide and the SUB protein interaction domain. These identified epitopes could be related to the protection and efficacy of the SUB-based vaccines, and new chimeras containing these protective epitopes could be designed using this new approach.

## 1. Introduction

Ticks are arthropod ectoparasites that transmit multiple pathogens, causing diseases in humans and animals [1,2]. Tick control is important because ticks are considered second only to mosquitoes as vectors of pathogens that cause diseases in humans and the most important vector of diseases in cattle. Ticks have numerous important effects on livestock worldwide, particularly in the northern hemisphere, primarily due to Lyme disease of which there have been 360,000 estimated cases in Europe [3] over the last two decades and approximately 300,000 reported cases in the United States each year [4]; other zoonotic tick–borne pathogens have also been reported [1,2,3,5,6]. About 80% of the world cattle population is under the risk of ticks and tick-borne diseases [7]. Traditional methods for the control of tick infestations have been based on the application of acaricides, repellents, and antibiotics. However, the impact of drug resistance and contamination on public and environmental health constitutes an important limitation for these practices [8,9]. Vaccination is an environmentally-friendly alternative for the control of tick infestations and the reduction in the capacity to transmit pathogens that impact human and animal health [10,11,12,13,14]. In Uganda, where the studied tick species *Rhipicephalus appendiculatus, R. decoloratus*, and *Amblyomma variegatum* are the most important ectoparasites infesting cattle, tick control using acaricides is a significant economic expense for livestock farmers, [15,16,17] and tick-borne diseases (TBD) affect cattle production with estimated losses of over USD 1.1 billion each year [18,19].

Subolesin (SUB) and Akirin (AKR) are orthologous proteins in ticks and insects that act as transcription factors affecting the expression of signal transduction and innate immune response genes [20,21,22]. The SUB tick-protective antigen was discovered in *Ixodes scapularis* [23], and vaccination with recombinant SUB showed an effective control of tick infestations by reducing their number, weight, oviposition, and pathogen infection [23,24,25]. SUB/AKR chimeric Q38 and Q41 antigens were designed in silico and have been used as vaccines for the control of ticks and other arthropod vectors reducing the risk of pathogen transmission to the infested host [26].

Recent studies have predicted the domains involved in protein–protein interactions [27,28]. In SUB/AKR, the alignment of the identified interaction domains with the protective epitopes of Q38 and Q41 chimeras covered more than 75% of the antigen protein sequence [29].

Novel approaches proposed the identification and combination of antigen-protective epitopes as well as the characterization of protein–protein interactions for vaccine development [29,30,31,32,33]. However, most of the studies are based on in silico or in vitro prediction of conserved epitopes. The basic approach of epitope mapping, used to study the interactions between antigens and antibodies, is now being extensively employed to map protective epitopes in immunized and protected individuals for the identification of relevant protein regions for vaccine design [20,34,35].

Quantum vaccinomics was proposed as a platform to target some of the challenges in vaccine development for the control of ticks, other ectoparasites, and infectious diseases [36]. This approach was based on the characterization of protein–protein interactions in the cell interactome and regulome in host–vector–pathogen interactions and the identification and characterization of protective epitopes in protein-interacting domains for the design and production of safe and more effective vaccines [35,36,37].

To further validate quantum vaccinomics as a methodology applied to the identification and characterization of protective and interacting epitopes of SUB, herein we mapped B-cell linear epitopes of SUB from three different tick species (*R. appendiculatus* as SUBra, *R. decoloratus* as SUBrd, and *A. variegatum* as SUBav) using serum samples from two cattle breeds immunized with SUB-based vaccines [25]. This methodology allowed the identification of epitopes recognized by IgG antibodies from immunized animals and confirmed their presence in SUB–protein interaction domains predicted in previous studies and in the Q38 chimera. These identified epitopes may be also related to the protection and efficacy of the SUB antigen and the chimeras containing its protective epitopes.

## 2. Materials and Methods

### 2.1. Serum Samples

Serum samples were obtained from a previous vaccination trail [25]. Five groups were selected for the study, and their composition is described in Figure 1. Bos indicus and *B. indicus* × *B. taurus* crossbred cattle breeds (4 animals per group) were vaccinated with (a) a cocktail of three SUB antigens from three tick species (*R. appendiculatus* (SUBra), *R. decoloratus* (SUBrd), and *A. variegatum* (SUBav)); (b) SUBra; and (c) adjuvant alone as a control. In this control group, samples from two animals of each cattle breed were selected and pooled. Serum samples from day 60 were selected because they showed the highest IgG antibody titers as it usually takes time for the immune system to produce highly effective antibodies [25,31].

### 2.2. Tick R. appendiculatus, R. decoloratus, and A. variegatum SUB Epitopes Microarray

The *R. appendiculatus, R. decoloratus*, and *A. variegatum* peptide SUB microarray elongated with neutral GSGSGSG linkers at the C- and N-terminus and translated into 471 different overlapping 15 amino acids (aa) peptides (peptide–peptide overlap of 14 aa) was printed in duplicate (942 peptide spots each array copy) at PEPperCHIP^®^ Immunoassay, PEPperPRINT, Germany. Serum samples from each group mentioned above were pooled (*n* = 4) and used to identify protective regions or epitopes in *R. appendiculatus, R. decoloratus,* and *A. variegatum* SUB (GenBank ID: ABA62331.1; AGI44619.1; QKX96321.1); then a high-resolution epitope mapping of SUB protein from the three tick species was performed. The peptide microarray was assembled in an incubation tray and blocked with 1% (*w/v*) bovine serum albumin (BSA) in PBS pH 7.4 with 0.005% (*v/v*) Tween-20 (PBST) for 30 min at room temperature (RT). After it was washed with PBST three times, the array was incubated with pooled sera diluted 1:500 in blocking solution overnight at 4 °C. The next day, it was washed again and the array was incubated with a monoclonal anti-bovine IgG antibody produced in mouse and previously stained with Mix-n-Stain™ CF™ 555 antibody labeling kit (Sigma-Aldrich, St. Louis, MO, USA) diluted 1:750 in blocking solution for 45 min at RT. The array was washed, dissembled from the tray, and dried with centrifugation for 1 min at 190× *g*. The resulting array was scanned with a GenePix personal 4100a microarray scanner (Molecular Devices, San José, CA, USA), and GenePix Array List (GAL) files supplied by a microarray slide manufacturer were used for image analysis. The median fluorescent signal intensity of each spot was extracted using MAPIX software (Molecular Devices, San José, CA, USA).

### 2.3. Data Analysis and Peptide Characterization

For data analysis, we used the intensity of the raw fluorescence signal corresponding to the median signal intensity subtracted from the median background intensity of each spot, then averaged across duplicate spots [38]. The resulting signals were normalized with a Z-Score [39,40], Z-Score = (intensity_P_—mean intensity_P1…Pn_)/SD_P1…Pn_, where *p* is any SUB peptide on the microarray and P1…Pn represent the aggregate measure of all the peptides. The heatmaps of IgG antibody binding to the peptides were visualized using the Z-Score heatmapper (http://www.heatmapper.ca/expression/ accessed on 13 August 2022), where peptides that showed Z-scores > 2 were considered significant reactive peptides. Z-ratios were used for multiple comparisons between peptides from different immunized groups with the control group and were calculated by taking the difference between the averages of the observed peptide Z-scores and dividing by the SD of all the peptide Z-score differences. A Z-ratio of ±1.96 is inferred as significant (*p* < 0.05). The analysis was focused on the epitopes with Z-ratio > 1.96 when comparing the peptide reactivity in one group to the same peptide in the control group (Appendix A).

Amino acid SUB protein sequences from the three tick species were aligned including *I. scapularis* SUB sequence (GenBank ID: AAV67031.1) and using Clustal Omega tool (https://www.ebi.ac.uk/Tools/msa/clustalo/ accessed on 13 August 2022). Sequences from Q38 (GenBank ID: JX193856) protective epitopes and SUB interaction domains were obtained from previously published results, and similar regions in the study sequences were identified [26,29].

## 3. Results

### 3.1. Rationale Sample Selection

The study was designed with the aim of identifying the protective epitopes of SUB from three different tick species recognized by IgG from two common cattle breeds in Uganda (*Bos indicus* and *B. indicus* × *B. taurus* crossbred) immunized with different combinations of these antigens. Sera from the groups were immunized with a cocktail of three tick species’ SUB antigens and SUBra, because SUB is a conserved protein with evidence of its function in protein–protein interactions [29]. Furthermore, SUBra showed high protection against three tick species from Uganda (*R. appendiculatus, R. decoloratus,* and *A. variegatum*), and its epitopes reactive with cattle IgGs may be involved in vaccine protection (Figure 1).

Based on the ELISA results of the previous study [25], we selected serum samples from day 60 of the experiment, which corresponds to the highest serum IgG antibody titers in the animals.

**Figure 1 vaccines-10-01327-f001:**
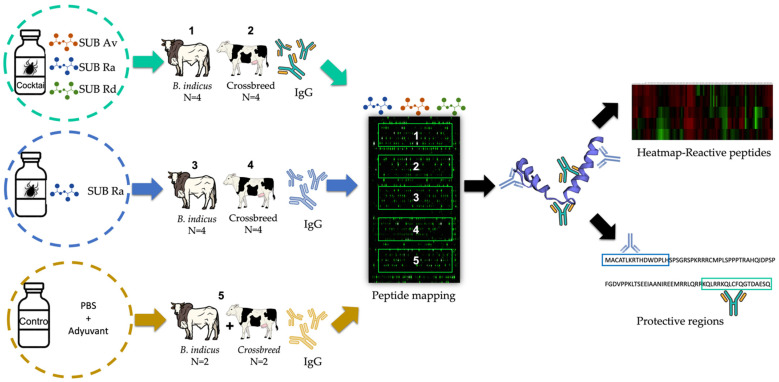
Experimental design of the study. Pooled serum samples used in the peptide mapping were from cattle breeds (*Bos indicus* and *B. indicus* × *B. taurus* crossbred) common in Uganda, immunized with SUB from *R. appendiculatus* (SUBra), *R. decoloratus* (SUBrd) and *A. variegatum* (SUBav), and combined in a cocktail, SUBra alone and a control group immunized with PBS. In the control group, serum samples from only two animals from each cattle breed were used. A peptide microarray was performed for the identification of SUBra, SUBrd, and SUBav reactive epitopes and protective amino acid regions recognized by IgG antibodies from the immunized groups.

### 3.2. Reactivity of Cattle Serum Samples against Subolesin Peptides

The scan peptide microarray containing aa sequences from the three tick species SUB in each copy was incubated with the corresponding pooled serum samples from each SUB-immunized group and the control (Figure 1). Reactive epitopes with a Z-score > 2 in at least one group are highlighted on the heatmap (Figure 2) in bright green, showing regions of highly reactive peptides recognized by the IgG of the immunized animals. However, some regions were also reactive for IgG from the control group immunized with PBS; these may be attributed to nonspecific antibody binding and were hence excluded from the regions of interest. The SUBav peptides 21–22 and 125–129; the SUBra peptides 19, 31, and 113–116; and the SUBrd peptides 111, 114, 115, and 117 (marked with red asterisk in Figure 2) showed a Z-score > 2 in the pooled sera from the control group.

The study identified common specific peptide sequences in the different SUB antigens which may be involved in the efficacy of these vaccine antigens (Figure 2). The overlapping peptides 142–152, 128–136, and 128–136 in SUBav, SUBra, and SUBrd, respectively, showed a Z-score > 2 in crossbred and *B. indicus* cattle vaccinated with SUBra (Table 1). Furthermore, peptide 110 in SUBav and peptide 94 in SUBra and SUBrd (Figure 2) with the sequence TSGLLSPVRRDQPLF in the three SUB antigens (Table 1) were significantly reactive to IgG antibodies from crossbred cattle vaccinated with SUBra, but only in SUBav and SUBrd. This peptide was also reactive to antibodies from *B. indicus* immunized with SUBra but was not reactive in the groups immunized with the SUB cocktail (Table 1).

The results also showed that all the reactive peptides recognized by the crossbred cattle immunized with SUBra and with a Z-score > 2 also showed a Z-ratio > 1.96 when compared with the control (Table 1).

Despite finding some reactive peptide regions with significant Z-scores in the SUB cocktail–immunized cattle, some of these overlapping peptides are also reactive in the control group corresponding to the sequence of amino acids 117-FTFRQVGLICERMMKERES-135 from SUBav, 108-LICERMMKERESKIRE-123 from SUBra, and 104-RQVGLICERMMKERESKIREE-124 from SUBrd (Table 1). In contrast, reactive peptides with a Z-score > 2 common to the control group did not appear in the groups of cattle immunized with SUBra.

In addition, cattle breeds immunized with the SUB cocktail showed only as overlapping reactive peptides with a Z-ratio > 1.96 the amino acid sequences: 72-RLQRRKQLCFQGAECSSPPEGC-93 from SUBav, 62-IREEMRRLQRRKQLC-76 and 111-ERMMKERESKIREEY-125 from SUBra, and 113-MMKERESKIREEYDH-127 from SUBrd.

### 3.3. Characterization of the Reactive SUB Peptides by Quantum Vaccinomics Approach and Vaccine Protection

An amino acid sequence alignment was performed to validate the quantum vaccinomics approach using the SUBav, SUVra, and SUBrd included in the study, as well as *I. scapularis* SUB. The peptides from the chimeric SUB/AKR antigen Q38 [26] and the SUB interaction domain [29] were highlighted.

The results of the alignment (Figure 3) showed that not all the peptides included in the Q38 chimera were recognized by the IgG antibodies of the immunized animals. The peptides of the Q38 antigen 1-MACATLKRTHDWDPLHSPNGRSPK-24 and 103-SPTGLSPGGLLSPVRRD-120 were not significantly recognized in the three tick species SUB by IgG antibodies from the SUB cocktail–immunized group (Figure 3A,B). Additionally, the Q38 peptides 1-MACATLKRTHDWDPLHSPNGRSPK-24, 48-PSPFGEVPPK-57, and 83-SSPLESGSPSATPPA-97 were also not significantly recognized by IgG from cattle groups immunized with SUBra (Figure 3C,D). However, peptide 83-SSPLESGSPSATPPA-97 was only recognized in SUBav by antibodies from cattle immunized with the SUB cocktail. The longest peptide from Q38 104-SPTGLSPGGLLSPVRRDQPLFTFRQVGLICERMMKERESQIRDEYDHVLSAKLAEQYDTFVKFTYDQIQKRFEGATPSYLS-184 was the only one partially recognized by the IgGs from all the immunized cattle groups (Figure 3A–D), being also part of the interaction domain of SUB shown in Figure 3.

## 4. Discussion

Quantum vaccinomics is a novel platform proposed for the identification and combination of antigen-protective epitopes for the development of more effective vaccines, even if applied to antigens that appear to be good candidates but offer low protection after challenges [29,36,41]. This approach focuses on the characterization of molecular interactions between host and pathogen using omics technologies [42] by identifying and characterizing immunogenic and protective epitopes as well as protein interaction domains as new strategies for the design and production of chimeric vaccine antigens [29,43,44].

In our study, immunogenic epitopes recognized by IgGs formed a crossbreed. *B. indicus* cattle immunized with the SUB cocktail and the SUBra were identified by epitope mapping, and the peptide overlapping regions were studied. The results are visualized in a heatmap (Figure 2) where reactive epitopes with a Z-score > 2 in at least one group are highlighted. The regions located between peptides 142–152, 128–136, and 128–136 in SUBav, SUBra, and SUBrd, respectively, seem very similar when these sequences are aligned (Figure 3); they showed a Z-score > 2 in crossbred and *B. indicus* cattle vaccinated with SUBra. These cattle groups, infested with *R. appendiculatus* and immunized with SUBra, showed vaccine efficacy from 47% in *B. indicus* to 90% in crossbred cattle in a previous publication [25]. Additionally, peptide 110 in SUBav and peptide 94 in SUBra and SUBrd (Figure 2) with the sequence TSGLSSPVHRDQPLF were significantly reactive to IgG from cattle vaccinated with SUBra. However, this peptide does not appear to be significant in any of the groups vaccinated with the SUB cocktail and may be a critical peptide in the protection provided by SUBra antigen–controlling infestations from the three tick species (Table 1).

The efficacy of previous vaccination studies demonstrated that crossbred cattle vaccinated with SUBra had higher efficacy in controlling tick infestations (90% efficacy for *R. appendiculatus*; 89% for *A. variegatum* and 51% for *R. decoloratus*) than *B. indicus* cattle vaccinated with the same antigen (47% efficacy for *R. appendiculatus* and 50% for *A. variegatum*) [25], suggesting a possible correlation between the reactive peptides identified by this cattle breed and the vaccine efficacy against these tick species. Some studies correlated the immunological protection with differences in immune response between cattle breeds [45]. These differences also affect tick resistance in the different breeds [46]. Therefore, the different efficacies obtained in the two cattle breeds could also be linked to differences in the immune response of each breed to the same antigen.

The overlapping reactive peptides (Z-score > 2) in the control group corresponding to the sequence of amino acids 117-FTFRQVGLICERMMKERES-135 from SUBav, 108-LICERMMKERESKIRE-123 from SUBra, and 104-RQVGLICERMMKERESKIREE-124 from SUBrd (Table 1) also showed significant Z-scores in the SUB cocktail–immunized cattle. This finding may be caused by nonspecific polyclonal antibody binding that could mitigate the real effect of the actual protective peptides [47]. These peptides did not appear as reactive in the groups of cattle immunized with SUBra, showing this group to have higher vaccine efficacy as previously reported [25]. Excluding the nonspecific antibody– recognized peptides and focusing on the immunoglobulin isotype could be requirements to achieve a more accurate and enhanced immune response that would allow the development of more effective vaccines. The development of vaccines is greatly aided by the identification of epitopes targeted by protective antibodies (protective B-cell epitopes). The knowledge provided by this approach could guide the design of subunit vaccines to include protective epitopes and exclude any epitopes that might induce autoimmune cross-reactive antibodies [48].

These findings encourage the identification of overlapping reactive epitopes to construct new chimeric vaccine antigens [49,50] as well as other applications related to diagnostics, design of individualized vaccines or as therapeutic targets [51]. The epitope structure is important for chimeric antigen vaccine development and confers the ability to direct the immune response to more effectively target epitopes [34,52]. The peptides analyzed in this study were linear B-cell epitopes, and most of the B-cell epitopes are located on the exposed parts of the antigen [53]. The capacity of the immune system to identify these surface antibody binding areas in the antigens sequence could affect vaccine efficacy [54]. Although the majority of peptide epitopes are discontinuous, an amino acid sequence containing all the residues of a discontinuous epitope is required for proper conformation of the contact residues. Therefore, B-cell epitopes define the contact residues and the conformation, which is determined by the three-dimensional fold of the contact residues [54]. A limitation of the methodology applied in this study is its restriction to linear epitopes; thus, even if a significant part of an epitope is a short linear peptide, this does not ensure that the peptide represents the entire epitope or that it does not require a different conformation [55]. Even short linear peptides may depend on their three-dimensional conformation for bioactivity [50]. Furthermore, the identification of protective B-cell epitopes may uncover or localize pathogenic functions, especially if the antibodies block targeted B-cell epitopes involved in host–pathogen interaction [56] or interaction with other proteins of the interactome network (protein–protein physical and functional interactions) [44,57], which makes this tool very useful in the control of ticks and tick-borne pathogens.

Nevertheless, results obtained from previous studies related to SUB vaccination suggest that this conserved antigen [58] could be used for the development of a universal vaccine for the control of various arthropod vectors [26,59]. The epitope mapping in this study identified SUB epitopes which, despite being conserved, showed differences among the different tick species by IgG recognition in different cattle breeds that may be implicated in vaccine efficacy for the control of *R. appendiculatus, R. decoloratus*, and *A. variegatum*. A SUB mapping was previously performed using sera from rabbits and sheep immunized with the recombinant tick and mosquito SUB, where differences in the linear B-cell epitopes identified between tick and mosquito SUB ortholog proteins were also found and could be attributed to the different secondary structure of the proteins [55]. In the present study, the differences in epitope recognition of the SUBra, SUBav, and SUBrd caused by the secondary structure of the protein cannot be excluded, and X-ray crystallography analysis could be performed in future experiments to identify epitope conformation [48]. However, the results encourage the design and development of multi-epitope vaccine antigens that could facilitate the presentation of appropriate B-cell epitopes to the host immune system and improve vaccine antigenicity.

Based on this approach, we compared and analyzed protective and interactive domains identified in previous studies [26,29]. For this purpose, an amino acid sequence alignment was performed (Figure 3) to validate if peptides identified by this approach may be part of interaction domains with an important role in the functionality of SUB [29] or Q38 protective peptides [26,60].

All the groups included in this study showed a similar peptide sequence recognized by IgG that is also a part of the interaction domain previously identified in SUB orthologs AKR1 and AKR2 and predicted for tick SUB [29,61], which represents 34% of the SUB sequence (Figure 3A–D). Furthermore, this interaction domain is 100% covered by the Q38 protective peptide with the amino acid sequence 123-LFTFRQVGLICERMMKERESQIRDEYDHVLSAKLAEQYDTFVKFTYDQIQKRFEGATPSYLS-184. Additionally, the cattle groups with the highest vaccine efficacy were *B. indicus* immunized with the SUB cocktail (92%) (Figure 3B) and crossbred cattle immunized with SUBra (90%) (Figure 3C) against *R. appendiculatus* infestations in both. In these animals, IgGs recognized epitopes from peptide 104-SPTGLSPGGLLSPVRRDQPLFTFRQVGLICERMMKERESQIRDEYDHVLSAKLAEQYDTFVKFTYDQIQKRFEGATPSYLS-184 (Z-ratio > 1.96; Figure 3), containing both Q38 epitopes and SUB–protein interaction domains. It is possible that in SUBra these domains are more susceptible to being recognized by antibodies due to their conformation being more effective in the control of this particular tick, but further studies should be carried out to confirm this. Considering that the SUB of the three tick species (SUBra, SUBrd, and SUBav) shared this reactive peptide domain recognized by IgGs from immunized animals of both cattle breeds (Figure 3), and since this domain has previously been identified as part of the SUB-interacting domain [29]—also part of the Q38 chimera which showed good vaccine efficacy [59,60]—this domain could be considered as potentially protective.

SUB is a conserved protein and plays a role in-cell interactome and regulome in response to pathogen infection and other biological processes in ticks [21,61]. A previous study characterized the functional evolution of SUB/AKR and their structure, protein–protein interactions, and function in different species and provided insights that these regulatory proteins have potential as vaccine antigens for the control of ectoparasite infestations and pathogen infection [61]. This research encouraged the development of new vaccine formulations by combining SUB/AKR with interacting proteins. The peptides identified in this study that are aligned with the SUB-interacting domain (Figure 3) could be a step forward in the development of SUB antigen–based vaccines by blocking this interaction domain. Interaction domain where is located the epitope core responsible for a physiological or biological function which may be shared by antibodies that target these overlapping epitopes [48,62]—one of the main objectives of quantum vaccinomics [36]. However, combining other mapping methods in future investigations may allow us to obtain complementary or supporting data, and further studies will be needed to confirm the protective efficacy of the identified epitopes.

Finally, SUB epitope mapping may be applied as a main tool in the quantum vaccinomics approach to the design of new chimeras. Such chimeras could facilitate the presentation of the B-cell epitopes and guide the host immune system toward production of protective antibodies [63,64] using combined reactive SUB epitopes found in these tick species. In this way, we will be able to obtain more effective multi-tick vaccines in different cattle breeds and/or include amino acid changes specific to the SUB of each tick species for a personalized vaccine design.

## 5. Conclusions

In conclusion, different epitopes in *R. appendiculatus, R. decoloratus*, and *A. variegatum* SUB proteins were identified by B-cell linear epitope mapping. A candidate-common protective domain located within the Q38 peptide sequence, and the SUB–protein interaction domain was significantly recognized by IgG antibodies from all the immunized groups. A multi-epitope vaccine could be designed to produce an enhanced and improved immune response for tick control using a quantum vaccinomics approach. However, further experiments will be needed to confirm if these identified epitopes are necessary and sufficient to confer better protection against infestations of multiple tick species in different cattle breeds.

## Figures and Tables

**Figure 2 vaccines-10-01327-f002:**
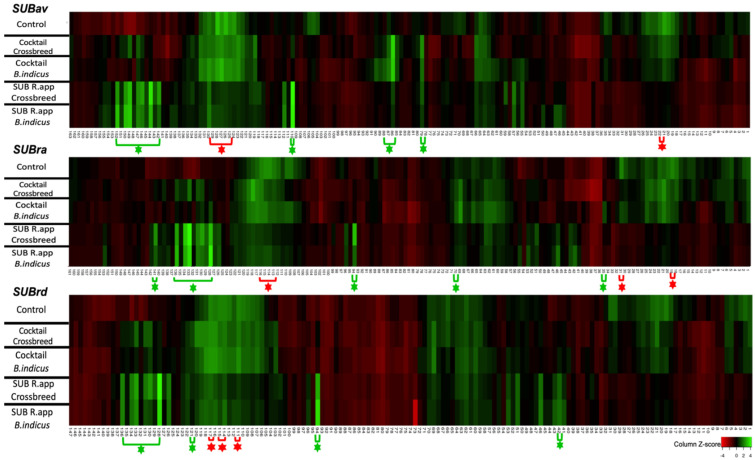
Heatmap of the IgG reactive epitopes in SUBav, SUBra, and SUBrd. Reactivity against peptides is indicated with Z-score, and possible epitope regions were identified with a green asterisk when the Z-score > 2. Red asterisks show peptide regions with significant reactivity with pooled sera from the control group.

**Figure 3 vaccines-10-01327-f003:**
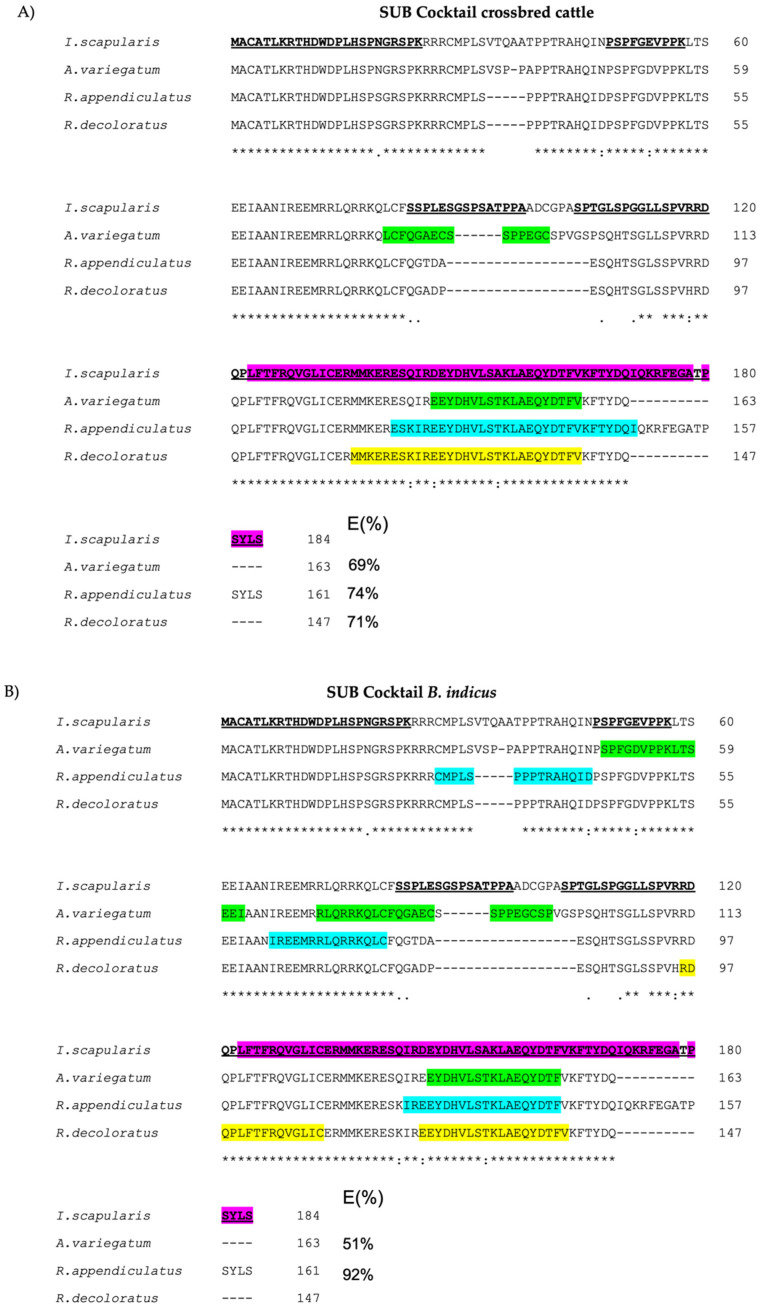
Alignment of SUB amino acid protein sequences from *I. scapularis*, *R. appendiculatus, R. decoloratus*, and *A. variegatum*. Reactive epitopes identified in the immunized groups (**A**) cocktail SUB in crossbred cattle, (**B**) cocktail SUB in *B. indicus*, (**C**) SUBra in crossbred cattle, and (**D**) SUBra in *B. indicus* with a Z-score significantly different (Z-ratio > 1.96) when compared with the control group are highlighted (green: *A. variegatum,* blue: *R. appendiculatus* and yellow: *R. decoloratus*. SUB/AKR interacting domain and protective epitopes included into Q38 chimera are shown. Vaccine efficacy (E%) of the SUB antigens with the different tick species as previously was described [25] is shown. Conserved amino acid residues between all sequences are indicated with asterisks (*) and numbers at the right denote the number of amino acid residue.

**Table 1 vaccines-10-01327-t001:** SUB overlapping peptides from three different tick species identified as reactive and protective in each of the study groups.

			Serum Groups				
Peptide n°	Tick Species	Peptide Sequence	Control	Cocktail Crossbred	Cocktail *B. indicus*	SUBra Crossbred	**SUBra *B. indicus***
20	A.v	^13^DPLHSPNGRSPKRRR^27^	x				
21	A.v	^14^PLHSPNGRSPKRRRC^28^	x				
55	A.v	^28^SPFGDVPPKLTSEEI^42^			†		
79	A.v	^72^RLQRRKQLCFQGAEC^86^		x †	†		
86	A.v	^79^LCFQGAECSSPPEGC^93^		x †	x †		
88	A.v	^81^FQGAECSSPPEGCSP^95^			†		
110	A.v	^103^TSGLLSPVRRDQPLF^117^				x †	x †
112	A.v	^105^GLLSPVRRDQPLFTF^119^					x
119	A.v	^112^RDQPLFTFRQVGLIC^126^		x			
123	A.v	^116^LFTFRQVGLICERMM^130^		x	x		
124	A.v	^117^FTFRQVGLICERMMK^131^	x	x	x		
125	A.v	^118^TFRQVGLICERMMKE^132^	x		x		
126	A.v	^119^FRQVGLICERMMKER^133^	x	x	x		
127	A.v	^120^RQVGLICERMMKERE^134^	x	x	x		
128	A.v	^121^QVGLICERMMKERES^135^	x	x	x		
129	A.v	^122^VGLICERMMKERESQ^136^			x		
130	A.v	^123^GLICERMMKERESQI^137^			x		x
132	A.v	^125^ICERMMKERESQIRE^139^			x		
142	A.v	^135^SQIREEYDHVLSTKL^149^				x †	
144	A.v	^137^IREEYDHVLSTKLAE^151^				x †	x †
146	A.v	^139^EEYDHVLSTKLAEQY^153^		†		x †	†
147	A.v	^140^EYDHVLSTKLAEQYD^154^		†	†	x †	x †
149	A.v	^142^DHVLSTKLAEQYDTF^156^		†	†	x †	x †
150	A.v	^143^HVLSTKLAEQYDTFV^157^		†		x †	x †
Peptide n°	Tick species	Peptide sequence	Control	Cocktail Crossbred	Cocktail *B. indicus*	SUBra Crossbred	SUBra *B. indicus*
152	A.v	^145^LSTKLAEQYDTFVKF^159^				x †	x †
19	R.a	^12^WDPLHSPSGRSPKRR^26^	x				
21	R.a	^14^PLHSPSGRSPKRRRC^28^		x	x		
31	R.a	^24^KRRRCMPLSPPPTRA^38^	x				
35	R.a	^28^CMPLSPPPTRAHQID^42^			†		
69	R.a	^62^IREEMRRLQRRKQLC^76^			x †		
94	R.a	^87^TSGLSSPVRRDQPLF^101^				x †	
113	R.a	^106^VGLICERMMKERESK^120^	x				
114	R.a	^107^GLICERMMKERESKI^121^	x				
115	R.a	^108^LICERMMKERESKIR^122^	x		x		
116	R.a	^109^ICERMMKERESKIRE^123^	x	x	x		
117	R.a	^110^CERMMKERESKIREE^124^		x	x		
118	R.a	^111^ERMMKERESKIREEY^125^		x †	x		
119	R.a	^112^RMMKERESKIREEYD^126^		x			
128	R.a	^121^IREEYDHVLSTKLAE^135^			†	x †	x †
130	R.a	^123^EEYDHVLSTKLAEQY^137^		†	†	x †	
131	R.a	^124^EYDHVLSTKLAEQYD^138^		†	†	x †	x †
132	R.a	^125^YDHVLSTKLAEQYDT^139^		†			
133	R.a	^126^DHVLSTKLAEQYDTF^140^		†	†	x †	x †
134	R.a	^127^HVLSTKLAEQYDTFV^141^		†		x †	x †
136	R.a	^129^LSTKLAEQYDTFVKF^143^				x †	x †
139	R.a	^132^KLAEQYDTFVKFTYD^146^		†			
141	R.a	^134^AEQYDTFVKFTYDQI^148^		†		x †	†
42	R.d	^35^PTRAHQIDPSPFGDV^49^					†
43	R.d	^36^TRAHQIDPSPFGDVP^50^					†
94	R.d	^87^TSGLSSPVHRDQPLF^101^				x †	x †
111	R.d	^104^RQVGLICERMMKERE^118^	x		x		
Peptide n°	Tick species	Peptide sequence	Control	Cocktail Crossbred	Cocktail *B. indicus*	SUBra Crossbred	SUBra *B. indicus*
114	R.d	^107^GLICERMMKERESKI^121^	x				
115	R.d	^108^LICERMMKERESKIR^122^	x		x		
116	R.d	^109^ICERMMKERESKIRE^123^		x	x		
117	R.d	^110^CERMMKERESKIREE^124^	x	x	x		
118	R.d	^111^ERMMKERESKIREEY^125^		x	x		
119	R.d	^112^RMMKERESKIREEYD^126^		x			
120	R.d	^113^MMKERESKIREEYDH^127^		x †			
128	R.d	^121^IREEYDHVLSTKLAE^135^				x †	x †
130	R.d	^123^EEYDHVLSTKLAEQY^137^		†	†	x †	
131	R.d	^124^EYDHVLSTKLAEQYD^138^		†	†	†	†
132	R.d	^125^YDHVLSTKLAEQYDT^139^		†			
133	R.d	^126^DHVLSTKLAEQYDTF^140^		†	†	x †	x †
134	R.d	^127^HVLSTKLAEQYDTFV^141^		†	†	†	†
136	R.d	^129^LSTKLAEQYDTFVKF^143^				x †	x †

A.v, *Amblyomma variegatum*; R.a, *R. appendiculatus*; R.d, *R. decoloratus*; x, reactive peptides with Z-score > 2. †, peptides that showed significant differences with Z-ratio > 1.96 when compared with the control group. Numbers in superscript indicate the position of the amino acid in the SUB protein sequence.

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
