# Peer review of "The Correlation between Subolesin-Reactive Epitopes and Vaccine Efficacy"

_vaccines, 2022, doi:10.3390/vaccines10081327_

Round 1
Reviewer 1 Report
The authors present the results of the epitope mapping of immunized cattle. Serum samples were used from two cattle breeds that were immunized with two different vaccines, a cocktail of SUB or SUBra alone, and also groups immunized with adjuvant alone were included. Epitope mapping was performed using peptide arrays covering the SUB from three different tick species, common in Uganda.
Tick vaccination is a very valuable method as it is environmentally friendly in comparison with the alternative methods of reduction of infestation and the consequences. The study is very important, as it possibly contributed to the knowledge on the efficiency of used vaccines, and further results based on this study can also influence novel vaccine design. My main comment to the authors would be that the Figures are not always self-explanatory, the presentation and also the Figure legends should be improved. Also the labels of the vaccines (or immunogen types) are not consistent throughout the publication, which would be very helpful for the reader.
My other suggestion would be that the authors split the Results and Discussion Section (for now, the Results are labeled with 3 and Conclusion with 5). This will also be helpful to distinguish between the results presented in this study and comparison with other studies.
A clearer explanation should be made in what way the results of the present study “validate quantum vaccinomics”, as authors suggests, possibly with a statistical approach.
Please find below a list of remarks, which I hope you will find helpful:
Line 30: second as vector? Please reword
Line 32: numerous health and economic effects, please reword, something like important effects. The supporting figures relating to damaging effects would be important in this context.
Line 67: I do not understand in what way the described approach contributes to safety of the vaccine
Line 88: Species’names should not be in italics in an italics-typed title
Line 95: the species and the databank entries are not in the correct order, please correct
Line 96: fullstop missing
Line 98: 1x PBS is PBS
Line 100: at least in my version, degree Celsius symbol does not look ok
Line 105: rpm should be expressed in g
Line 116: were visualized
Line 140: its protective epitopes may be associated with vaccine protection, awkward sentence, please reword
Line 145: the immunization regime is not consistently described in the figure and the text. The figure indicates that the cocktail contained SUBra, av and rd, and the legend indicates the SUBap, av and rd. Further, the line 149 describes the mapping of SUBap, av and rd, and the Figure SUB ra, av and rd. Which is of course OK if the SUBra is the same as SUBap, but can you please explain this in the introductory paragraphs and stick to the same labels throughout the text.
Line 159: awkward sentence, please reword to: and were hence excluded from regions of interest, or similar
Footnote to Table 1: the control group
The legend to figure 2 at the right corner, above the line 164 is poorly visible – please enhance
Table 1 title: reactive peptides have very well been identified, but what indicates that these peptides when used for vaccination are protective? Please explain.
Line 200: (referring to Table 1): listed peptides are of different sequences than in the Table (the number of residues is different) – please explain the text.
Figure 3: carries labels a, b, c and d, which are not indicated in the figure legend, and they are not self-explanatory, please be more concise in the description of the immunized population and the immunogen. The boxed element at the bottom should be better labelled: what does the „recognized in SUB xxx” mean, recognized in this peptide microarray? I think there is an intended citation of reference no. 22, which is not commented. I suppose E% refers to the vaccine efficacy as described in the reference 22?
Do the numbers at the right of each amino acid sequence denote the number of residues? Are these really aligned sequences of each protein?
The resolution of the Figure could also be improved.
Lines 250-262: please list the sequential numbers of the peptides listed
Line 259: the peptide is difficult to identify as the identification numbers are missing
Line 267: decoloratus
Line 277: unjustified hyphenated presentation of the sequence
Line 277: this peptide does not appear in Table 1
Line 291: can you please explain what is the background of the reasoning of that the reactive peptides are also protective in vaccination? Please explain.
Line 282: poor wording: recognized peptide region, reactive to all groups may be mainly involved in the protection conferred by the SUB from these 3 tick species. What does reaction to all group mean? Where is the evidence that the antibodies to this peptide offer protection?
Reviewer 2 Report
This manuscript "Correlation between Subolesin reactive epitopes and vaccine efficacy" is a very well planned and written manuscript. The data is sound and could be an interesting read for the readers. I will like to do changes in the manuscript before acceptance.
1. The abstract though very good but is more theoretical. Therefore need to addition of numerical values and numbers of significant results.
2. Some figures quality may be improved like Figure 3.
3. I am not seeing any comparative analysis of the results with literature?
4. Limitations of the study must be mentioned in the discussion or conclusion section.
Round 2
Reviewer 1 Report
The authors have dilligently responded to reviewer's questions and corrected the ambiguities. The manuscript was restructured and now more clearly presents the data, the related background studies, and the prospective efforts to gain better insights into the activity of the used vaccines. The quality of figures was improved as well. Taking into account the extreme value of the topic of the research for society, I would like to recommend the manuscript for publication.